# Visual attention is available at a task-relevant location rapidly after a saccade

Tao Yao[1]*, Madhura Ketkar[1,2], Stefan Treue[1,3,4], B Suresh Krishna[1]*

[1]Cognitive Neuroscience Laboratory, German Primate Center, Goettingen, Germany; [2]European Neuroscience Institute, Goettingen, Germany; [3]Bernstein Center for Computational Neuroscience, Goettingen, Germany; [4]Faculty of Biology and Psychology, Goettingen University, Goettingen, Germany

**Abstract** Maintaining attention at a task-relevant spatial location while making eye-movements necessitates a rapid, saccade-synchronized shift of attentional modulation from the neuronal population representing the task-relevant location before the saccade to the one representing it after the saccade. Currently, the precise time at which spatial attention becomes fully allocated to the task-relevant location after the saccade remains unclear. Using a fine-grained temporal analysis of human peri-saccadic detection performance in an attention task, we show that spatial attention is fully available at the task-relevant location within 30 milliseconds after the saccade. Subjects tracked the attentional target veridically throughout our task: i.e. they almost never responded to non-target stimuli. Spatial attention and saccadic processing therefore co-ordinate well to ensure that relevant locations are attentionally enhanced soon after the beginning of each eye fixation.

**\*For correspondence:** taoyao@dpz.eu (TY); skrishna@dpz.eu (BSK)

**Competing interests:** The authors declare that no competing interests exist.

## Introduction

The processing of vision and visuospatial attention mostly proceeds via retinotopic representations in the brain (*Cavanagh et al., 2010*; *Wurtz, 2008*). Since each saccadic eye-movement leads to a change in the retinotopic representation of the visual scene, maintaining attention at a task-relevant spatial location across a saccade necessitates a rapid, saccade-synchronized shift of attentional modulation from the neuronal population representing the task-relevant location before the saccade to the one representing it after the saccade (*Marino and Mazer, 2016*; *Yao et al., 2016*). Currently, perceptual measurements in humans suggest a neuronal attention shift that starts before the saccade and continues after the saccade ends (*Rolfs et al., 2011*; *Szinte et al., 2015*; *Golomb et al., 2008*, *2010a*, *2011*; *Jonikaitis et al., 2013*). However, because these previous measurements used coarse temporal sampling and/or long-duration attentional probes, the precise time at which spatial attention becomes fully allocated to the task-relevant location after the saccade remains unclear. Here, using a fine-grained temporal analysis of human peri-saccadic detection performance in an attention task, we show that spatial attention is fully available at the task-relevant location within 30 milliseconds after the saccade. This rapid post-saccadic recovery of performance in our attention task indicates that retinotopic attentional shifts occur within the time required to recover from saccadic suppression of vision. Subjects almost never responded to the distractor change, indicating that they tracked the attentional target veridically throughout the task. Spatial attention and saccadic processing therefore co-ordinate well to ensure that relevant locations are attentionally enhanced soon after the beginning of each eye fixation.

**eLife digest** When we look at a scene, our gaze does not move continuously across it. Instead, our eyes move discontinuously, shifting gaze rapidly from point to point to focus on different locations in the scene. These eye movements are known as saccades, and during them the brain temporarily and selectively stops processing visual information.

In the brain, a particular area of a scene is represented by different neurons before and after a saccade. Paying attention to a relevant location in a scene across an eye movement therefore requires the brain to shift its attentional effects from the neurons that represented that location in the scene before the saccade to the set of neurons that do so after the saccade. Ideally, this shift should happen rapidly and be synchronized with the eye movement.

Exactly how long it takes for attention to emerge at a relevant location after a saccade was not clear because attention had not been recorded on a fine enough time-scale immediately after an eye movement. Yao et al. have now addressed this issue in a series of experiments that asked volunteers to focus their eyes on a fixed point. The volunteers had to follow the point with their eyes as it jumped to a new location, and at the same time had to look out for a change in the movement of a pattern of random dots.

The results reveal that attention is fully available at the relevant location within 30 milliseconds after the saccade. In fact, the 30-millisecond delay in the emergence of attention matches the period during which vision is suppressed during a saccade. Thus, the change in the brain's focus of attention coordinates with the saccadic eye movement to ensure that attention can be fixed on a relevant location as soon as possible after the eye movement ends. More studies are now needed to investigate how the brain coordinates its attention and eye-movement processes to synchronize the shift in attention with the eye movement.

## Results and discussion

We measured human peri-saccadic attentional allocation by combining an endogenous spatial attention task with a visually-guided saccade. Human subjects had to make a saccade to follow a fixation point when it jumped to a new location, and concurrently, pay attention throughout the trial to a target moving random-dot pattern (RDP) presented eccentrically among three or five physically similar distractor RDPs (*Figure 1*, and 'Materials and methods'). We measured the subjects' attentional allocation by their ability to detect a brief (23.5 ms) change in target motion, while ignoring similar changes in the distractors. The target and distractor changes occurred at different times around the saccade, allowing us to report for the first time, peri-saccadic performance in an attention task with fine-grained temporal precision. The intervening saccade poses a challenge for the attentional system, because due to the retinotopic shift of the target location across the saccade, the attentional system needs to shift its modulatory influence from the neuronal population representing the target before the saccade to the neuronal population representing the target after the saccade. By using a fixed timing and location for the fixation point jump, we could isolate the dynamics of this attentional remapping process and minimize its interaction with the dynamics of attentional allocation to other exogenous visual events. We therefore made the saccade spatially and temporally predictable by having the fixation point jump at the same time and to the same location on each trial so that the subject could best focus on the target pattern.

In Experiment 1, we looked at the peri-saccadic performance of 8 subjects (pooled data in *Figure 2A*, individual subject-data in *Figure 2—figure supplement 1*). At times well before and well after the saccade, subjects almost always detected the target change and their performance was near 100%. Performance began to drop around the time the fixation point jumped (dashed vertical line in *Figure 2A*), as expected from the previously reported diversion of pre-saccadic attentional resources towards the saccade task (*Deubel and Schneider, 1996*; *Montagnini and Castet, 2007*; *Hoffman and Subramaniam, 1995*) and the post-saccadic retinotopic location (*Szinte et al., 2015*). The performance then dropped steeply right before the saccade, as expected from the drop in visual sensitivity due to saccadic suppression (*Diamond et al., 2000*; *Dorr and Bex, 2013*; *McConkie and Loschky, 2002*). Importantly, our data show (for the first time in an attention task, to

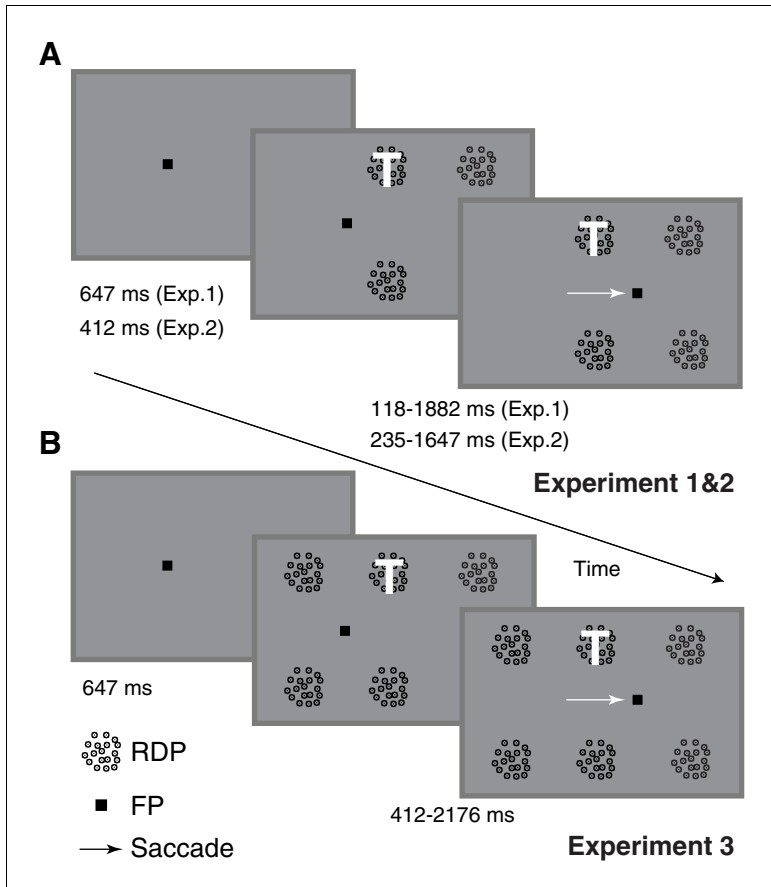

**Figure 1.** Task-design and timing. Human subjects performed a task that involved attending to a target (marked with a white T and always at the same location) presented as one among four (**A**) or six (**B**) moving RDPs while also making a visually guided saccade if the fixation point jumped seven degrees to the right (1235 ms after trial onset). The subjects were instructed to respond with a key-press when the target RDP briefly changed speed and direction, but to ignore similar changes in any of the remaining RDPs. Target and distractor changes occurred at different times around the saccade, enabling the measurement of peri-saccadic performance in this attention task. Two different task-difficulties were used in Experiment 2, while six RDPs were used in Experiment 3 instead of four. There were also minor differences in timing between the three tasks.

our knowledge) that performance recovered back to baseline within 30 ms of saccade offset (*Figure 2A*). The rapid post-saccadic recovery of performance indicates that attention is allocated to the task-relevant location within 30 ms after the saccade ends. The rapid time-course of recovery resembles that previously shown for saccadic suppression of visual performance in tasks where visual sensitivity was probed around a saccade using a briefly flashed change, but without any requirement to maintain attention on a target across a saccade (*Diamond et al., 2000*; *Dorr and Bex, 2013*; *McConkie and Loschky, 2002*). This suggests that while resumption of visual function after a saccade is constrained by the recovery from saccadic suppression (*Krekelberg, 2010*), the peri-saccadic attentional shift necessitated by retinotopic visual processing does not impose an additional temporal cost on this recovery. The rapid post-saccadic attentional availability at the target location that we infer from our data is consistent with the only physiological data on this issue: in a mental curve-tracing task similar to ours with a fixed attentional target, attentional effects in monkey V1 emerge approximately 80 ms after the end of the saccade (*Khayat et al., 2004*). Given an onset latency of approximately 30 to 50 ms in monkey V1, MT and LIP (*Khayat et al., 2004*; *Bair et al., 2002*; *Bisley et al., 2004*), a change occurring 30 ms after saccade offset would reach the visual cortex at approximately the time when its neurons representing the target after the saccade are attentionally enhanced.

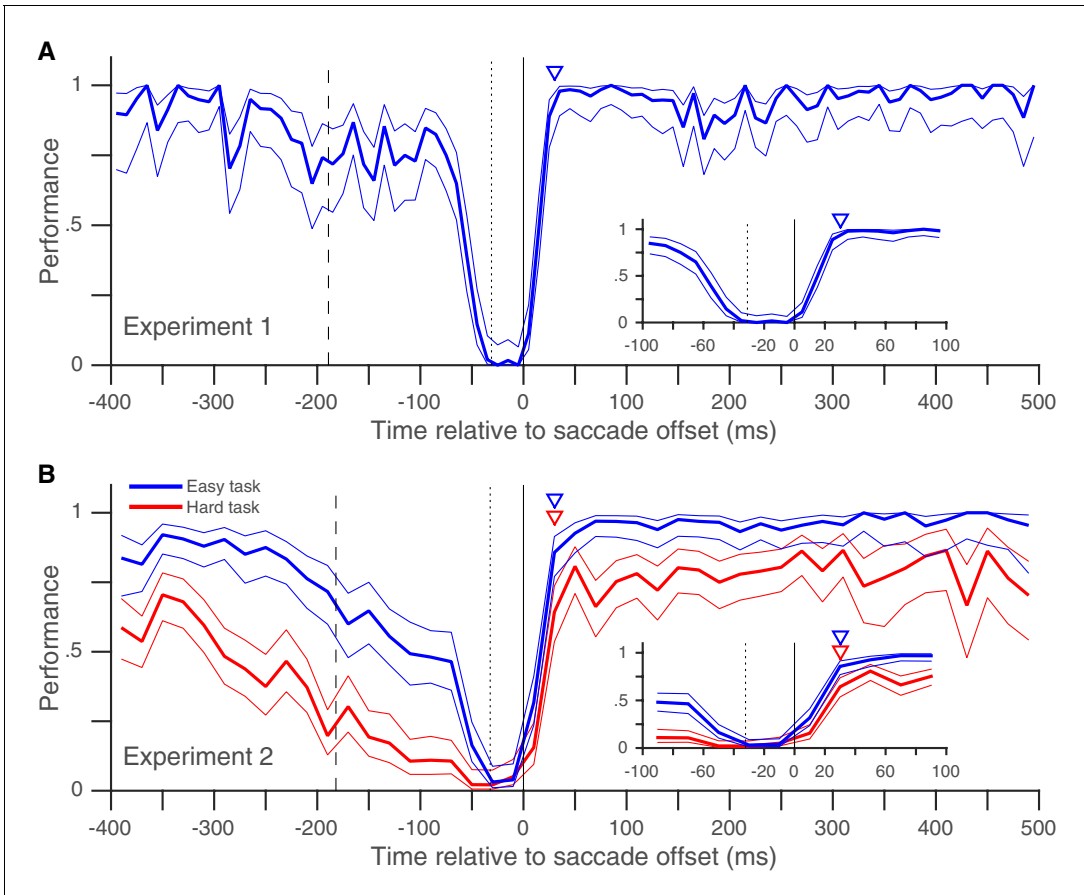

**Figure 2.** Rapid post-saccadic recovery of performance. (**A**) Detection-performance (hit-rate) of motion-direction drops around the time of the saccade and recovers within 30 ms after the saccade. The figure shows the mean detection-performance (and 95% confidence bands) for all trials pooled over 8 subjects calculated in non-overlapping 10 ms time-bins of the abscissa (time of target-change relative to saccade offset). The inset shows the same data, focusing on the time between −100 and 100 ms. Data from individual subjects show little inter-individual variability in the time-course of recovery (*Figure 2—figure supplement 1*). The triangle indicates the earliest time (30 ms) at which performance is statistically indistinguishable from that over the 100 to 500 ms time-period (using Boschloo's exact test; see 'Materials and methods'). The dashed vertical line indicates the mean time of fixation-point offset and the stippled vertical line indicates the mean saccade onset time. See also *Figure 2—figure supplement 1* and *3*. (**B**) Similar results were obtained when two different task-difficulties were used (data pooled over 5 subjects). The data from the higher-difficulty task (in red) show that the rapid recovery is not an artifact of a ceiling effect on performance. Data plotted using 20 ms time-bins. Figure conventions as in *Figure 2A*. See also *Figure 2—figure supplement 2* for data from individual subjects. *Figure 2—figure supplement 4* and *5* replot the same data as in *Figure 2A and B* and in the same format, but *Figure 2—figure supplement 4* uses the time of target-change relative to saccade onset and *Figure 2—figure supplement 5* only includes trials where a fixation window of 0.5° was used (see corresponding legends for details).

The following source data and figure supplements are available for figure 2:

**Source data 1.** Data plotted in *Figure 2A* and *Figure 2—figure supplement 1*.

**Source data 2.** Data plotted in *Figure 2B* and *Figure 2—figure supplement 2*.

**Source data 3.** Data plotted in *Figure 2—figure supplement 3*.

**Source data 4.** Data plotted in *Figure 2—figure supplement 4*.

**Source data 5.** Data plotted in *Figure 2—figure supplement 4*.

**Figure supplement 1.** Individual subjects – rapid post-saccadic recovery of performance.

**Figure supplement 2.** Individual subjects – rapid post-saccadic recovery of performance for two task difficulties.

*Figure 2 continued on next page*

*Figure 2 continued*

**Figure supplement 3.** Results from Experiment 3, where distractor changes are more numerous and more salient also show rapid post-saccadic recovery of performance (within 30 ms), and no evidence for post-saccadic retinotopic persistence or pre-saccadic predictive shifts.

**Figure supplement 4.** Post-saccadic recovery of performance plotted relative to saccade onset.

**Figure supplement 5.** Post-saccadic recovery of performance plotted with a smaller fixation window.

It is possible that though we report a rapid recovery in Experiment 1, the true recovery was actually slower, but was masked by the fact that performance had already reached its maximum value of 100% within 30 ms of saccade offset. We therefore performed a similar experiment (Experiment 2) with two task difficulties, where peak performance on the harder task was clearly below 100% (*Figure 2B*). Once again, performance recovered to baseline levels within 30 ms of saccade offset in both the easier and the harder task, indicating that our estimate of a rapid recovery time for performance was genuine and not an artifact due to a ceiling effect. The recovery time-course after the saccade also did not seem to depend on saccade latency (*Figure 3*). Very similar performance was observed when we grouped trials based on saccade latency into three groups: putative predictive saccades (latencies from 0 to 75 ms), express saccades (latencies from 75 to 125 ms) and regular-latency saccades (latencies from 125 to 250 ms). This indicates that though various differences between these different kinds of saccades have been noted and these different kinds of saccades have been speculated to arise via different neural pathways (*Bronstein and Kennard, 1987*; *Chen et al., 2013*; *Cotti et al., 2009*; *Deubel, 1995*; *Gaymard et al., 1998*; *Pierrot-Deseilligny et al., 2002*; *Shelhamer and Joiner, 2003*), peri-saccadic attentional shifts seem to proceed with a similar time-course in each case.

If the peri-saccadic attentional shift is not temporally well-synchronized with the saccade, attention will be peri-saccadically allocated to irrelevant spatial locations. In fact, prior findings measuring discrimination performance for attentional probes at different locations suggest that by about 75 ms before the saccade, attentional enhancement could be seen at the 'post-saccadic' retinotopic location (which would be the wrong pre-saccadic spatial location) (*Rolfs et al., 2011*). Other studies

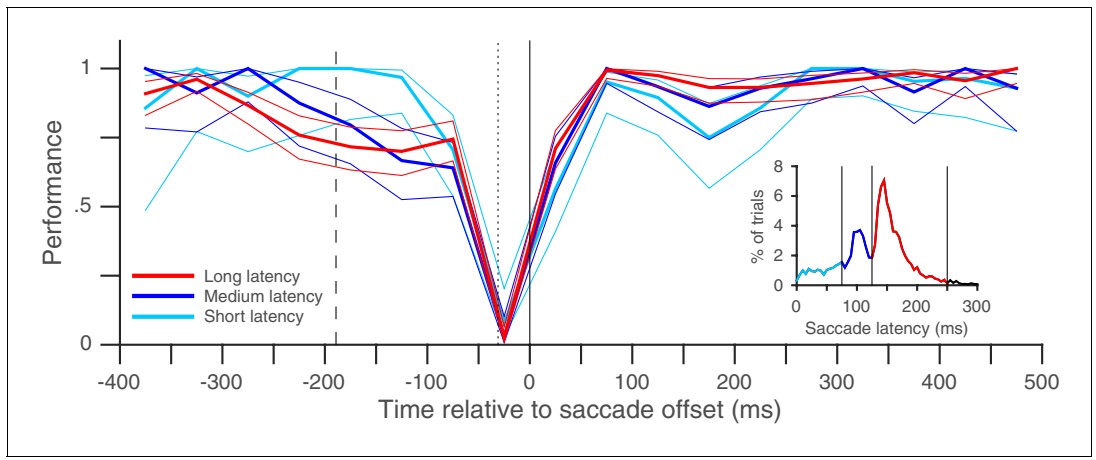

**Figure 3.** Rapid post-saccadic performance recovery is independent of saccade latency. The time-course of recovery was indistinguishable for saccades in three different latency ranges in the same dataset used in *Figure 2A* (8 subjects, color coding in inset): 0–75 ms (predictive saccades), 75–125 ms (express saccades), 125–250 ms (regular-latency saccades). The inset plots the pooled saccade latency distribution. Figure conventions as in *Figure 2A*, except that non-overlapping 50 ms time-bins were used.

The following source data is available for figure 3:

**Source data 1.** Data plotted in *Figure 3*.

report that after the saccade, attention stays at the pre-saccadic retinotopic location (which would be the wrong post-saccadic spatial location) for up to 100 ms after the saccade (*Golomb et al., 2008*, *2010b*). The predictive emergence of attention is consistent with single-neuron data from monkeys showing predictive responses in different attentional control areas of the brain (*Wurtz, 2008*; *Duhamel et al., 1992*), while imaging data from humans have been presented as evidence for persistent retinotopic neural activity (*Golomb et al., 2010a*). Results from a more recent detailed study indicate that peri-saccadic attentional spread and dynamics may show complex patterns: patterns consistent with predictive shifts, transient retinotopic persistence as well as rapid post-saccadic availability of attention at the task-relevant location were seen (*Jonikaitis et al., 2013*). In our data, we found no effect of predictive or delayed shifts on the rate of responding to distractor changes (false-positives). In both Experiments 1 and 2, overall, subjects responded to a distractor change on only 2.2% and 2% of trials respectively. Distractor changes occurred either at the distractor vertically below the target (a control) or at the distractor to the right of the target (that tested post-saccadic retinotopic persistence of the pre-saccadic attentional focus). In the time interval immediately after the saccade (0–150 ms), the data from both Experiments showed no statistically significant increase in the rate of false-positives due to retinotopic persistence (compared to the control location; all p-values>0.16, Boschloo's test; *Supplementary file 1A*). An additional experiment (Experiment 3; *Figure 2—figure supplement 3* and *Supplementary file 1B*) where we changed the task design to test both pre-saccadic predictive shifts and post-saccadic retinotopic persistence (while making distractor changes more numerous and salient) also led to a false-positive rate of less than 1.4% and no evidence for an effect of either predictive shifts or retinotopic persistence on the false-positive rate. Subjects thus tracked the attentional target veridically throughout our task, and the peri-saccadic spread of attention to irrelevant spatial locations reported in previous studies does not seem to have any manifest effects in our task. One important difference between our task and previous tasks was that we included only one attended location within each trial, and stimuli at all other locations were distractors that the subject had to ignore. In contrast, the previous tasks required subjects to report a probe stimulus that could appear at any of the stimulus locations. There were no distractor stimuli, and attention was instead manipulated by using a dual-task (*Rolfs et al., 2011*; *Golomb et al., 2008*, *2010b*) or using an exogenous cue (*Jonikaitis et al., 2013*). The fact that all stimulus locations on each trial were potential targets in the previous studies may have led the subjects to adopt a different attentional-set compared to the subjects in our study. Alternatively, the previous results may have reflected only an attentional effect on probe visibility, while the results in our task may additionally reflect the effect of attention on distractor filtering. In current theoretical accounts of attention (*Eckstein et al., 2009*; *Lu and Dosher, 1998*), the effects of attention on distractor filtering and probe visibility correspond to the distinct effects of attention on selection/weighting and sensory signal enhancement respectively. In this scenario, distractor filtering due to the attentional selection/weighting of sensory signals across the visual field is well-synchronized to the saccade and therefore does not spread to irrelevant spatial locations. In contrast, attentional signal enhancement, but not distractor filtering, is influenced by the predictive shifts and post-saccadic retinotopic persistence of attentional modulation in the brain. As a result, in the previous tasks without a distractor filtering component, the perceptual visibility of probes at irrelevant locations was improved. In our task, any enhanced sensory signal from distractor locations would continue to be down-weighted and filtered out and the subjects would not respond to them. We emphasize that this is only one plausible explanation, and theoretical models of attention are sufficiently complex and flexible to admit alternative explanations. Even more generally, the observed differences could be a result of task-dependent (or even entirely different) attentional mechanisms operating in the different tasks. Extensive measurements and model-testing will be necessary to disambiguate the different possibilities.

Our data represent an important advance in the ongoing discussion about the shifts of spatial attention around the time of a saccade (*Cavanagh et al., 2010*; *Marino and Mazer, 2016*; *Rolfs and Szinte, 2016*; *Mayo and Sommer, 2010*; *Melcher, 2010*). We provide the first temporally fine-grained measurements of detection performance in an attention task in the critical immediate post-saccadic period (0 to 100 ms following saccade offset). Our data show that performance fully recovers soon after the end of the saccade, indicating that the correct stimulus is attended to during this immediate post-sacadic period when visual sensitivity is known to be highest (*Ibbotson and Krekelberg, 2011*). The rapid time-course of recovery resembles the time-course previously shown for

the recovery of visual function from saccadic suppression, suggesting that the retinotopic attentional shift does not impose an additional temporal cost on the resumption of visual function after a saccade. Further, our data indicate that under our task conditions, subjects very rarely confuse a distractor stimulus for the target. Spatial attention and saccadic execution thus appear to co-ordinate well to ensure that relevant objects are attentionally enhanced soon after the beginning of each eye fixation. These findings are likely to lead to a much better understanding of the impact of peri-saccadic changes in neural activity on visual processing.

## Materials and methods

We measured peri-saccadic attentional allocation by combining a spatial attention task with a visually-guided saccade. We asked human observers to make a saccade to a visual target, and within the same trial, also report a speed and direction change in a target moving random-dot pattern (RDP), while ignoring a similar change in one of the simultaneously-presented distractor RDPs. The target and distractor changes occurred at different times around the saccade, allowing us to measure peri-saccadic attentional performance with fine-grained temporal precision.

### Human subjects

10 subjects (4 males, 6 females, ages 21–30 years) participated in the study, including two of the authors (MK and TY). 8 of the subjects (excluding the two authors) were naïve to the purpose of the Experiment. 8, 5 and 4 subjects participated in Experiments 1, 2 and 3 respectively; of these, 3 subjects (including the author MK) participated in all 3 Experiments. All subjects were right-handed and reported normal or corrected to normal vision. All naive participants received monetary compensation for each session. Each subject started the experiment with a training session to become familiar with the tasks. The experiments were performed in several blocks over one or two days. Subjects were given verbal and written instructions about the task. The study was performed in accordance with institutional guidelines for experiments with humans, adhered to the principles of the Declaration of Helsinki and was approved by the Ethics Committee of the Georg-Elias-Müller-Institute of Psychology, University of Göttingen. Each subject gave informed written consent prior to participating in the study.

### Apparatus

Subjects were seated in a dimly lit room at a viewing distance of 57 cm from the screen with their head resting on a chin and forehead-rest. The only light source in the room was the light from the display monitor. A computer keyboard was used for recording subject responses. All aspects of the experiment were controlled by custom software running on an Apple Macintosh computer. The eye-position was monitored by an infra-red video-based eye-tracker (iView X software running on an SMI Hi-Speed 1250 tracker, SMI GmbH, Germany) at 1000 Hz. The stimuli were displayed on a 1600 by 1200 pixels (40 by 30°) CRT monitor with a fresh rate of 85 Hz. The display background was always grey (40 cd/m$^2$), and all the visual stimuli were black (0.7 cd/m$^2$).

### Task design

We describe Experiment 1 first. Experiments 2 (*Figure 2B*) and 3 (*Figure 2—figure supplement 3*) are variants of Experiment 1.

#### Experiment 1

Each trial was started by the subject pressing the space-bar. A fixation point appeared on the screen and subjects maintained their gaze within 2° of this point. The subjects concurrently performed a spatial attention task and a saccade task on each trial: they were instructed to pay attention to the target RDP and make a saccade if the fixation point jumped to a new location. For the spatial attention task, after 647 ms of fixation, four circular moving random-dot patterns (RDPs: each presented within a circular aperture of 2° radius, with dots moving upwards with a speed of 8° of visual angle per second; dot density = 10 dots per deg$^2$), were displayed on the screen. Individual RDP dot size was 0.15 × 0.15°. The subjects were instructed that the RDP at a pre-specified location (3.5° to the right and 4° above the fixation point: *Figure 1*) was the target: they had to pay attention to that

stimulus throughout the trial in order to respond by pressing the downward-arrow key within 600 ms when they detected a brief 2-frame (23.5 ms) speed and direction-change in the target RDP. For these two frames, the RDP dots moved faster at 16° per second and horizontally either to the left or the right, and then resumed motion with the original speed and direction. Any changes in the distractor RDPs were to be ignored. The median reaction-time was 324 ms. The second RDP was placed 7° to the right of the target RDP so that post-saccadically, it had the same retinotopic location as the target RDP did pre-saccadically. The other two RDPs were placed at corresponding locations in the lower hemifield. A target RDP change occurred on about 90% of the trials and between 118 to 1882 ms after RDP onset; the remaining trials were catch trials and no change occurred. A distractor change occurred before the target change on about 39% of trials, over a similar range of times but at least 400 ms before the target change: the subject had to ignore these changes. Only one target change and possibly also one distractor change occurred on each trial. The distractor change could occur either at the RDP to the right of the target (with a post-saccadic retinotopic location identical to the target's pre-saccadic retinotopic location) or the RDP below the target. For the saccade task, the fixation point jumped to a new location 7° horizontally to the right of the fixation point 1235 ms after the fixation point appeared. The subjects had to make a single saccade within 553 ms of the fixation point jump to fixate the saccade target location and then maintain their gaze within 2° of the saccade target for the remainder of the trial. Most saccades occurred with a much shorter latency (*Figure 3*). The use of a predictable time at which the fixation point jumped was advantageous because subjects could focus their attention better on the target RDP without worrying about the temporal uncertainty about when the fixation point would jump. Trials were terminated when the subject pressed the downward arrow key, broke fixation or failed to press a key within 600 ms of a change in the target RDP. Subjects received no other feedback about trial outcome. The use of a 2° fixation window during the two fixation periods was not critical. We also obtained similar results when using a narrower fixation window of 0.5°: we ensured that the eye did not deviate by more than 0.5° from the median horizontal and vertical eye-position during fixation on each trial (*Figure 2—figure supplement 5*). Using the median eye-position compensates for across-trial drifts in calibration and is based on the standard calibration assumption that normal-viewing subjects will foveate a visual target when asked to fixate on it and therefore, their eye-position variability will be centered on the fixated location.

## Experiment 2

This was similar to Experiment 1, with the following key differences. Two task difficulties were used and the change involved only a motion direction change, without a speed change. The two task difficulties were created by using two magnitudes of direction change for each subject; these magnitudes were chosen in a separate calibration session,where the overall detection performance was estimated for nine direction-change magnitudes between 20 and 90°. The calibration session used a fixation task similar to the task in Experiment 2 except that no saccade was required. The direction-changes that led to approximately 70% and 90% detection performance were chosen for Experiment 2. Across subjects, the direction-change varied between 35 and 60° for the hard task and between 50 and 90° for the easier task. Also, to make more of the target changes occur peri-saccadically, the timing of the task was slightly modified so that the RDPs came on at 412 ms after fixation point onset (*Figure 1*), and the target motion change occurred from 235 ms to 1647 ms after RDP onset; approximately 27% of trials had a distractor change before the target-change over a similar time-frame (118 ms to 1224 ms, with the same constraint of a 400 ms separation from the target-change as in Experiment 1). About 7% of trials were catch trials.

## Experiment 3

This was also similar to Experiment 1, except that we used 2 additional distractor RDPs, giving a total of 6 RDPs instead of 4. One of the additional RDPs was placed seven degrees to the left of the target RDP, which is the location to which attention would be expected to predictively switch just before the saccade. The other RDP was placed eight degrees below this RDP, in line with the other RDPs in the lower hemifield. Further, to make the distractor changes more salient and improve the chances of a false-positive, the speed now increased during the motion change from 4 to 32° per second (instead of 8 to 16° per second in Experiment 1); the direction-change remained at 90°

(vertically upward to horizontal towards the left or the right). The range of target change times was slightly delayed compared to Experiments 1 and 2 so that a distractor change could occur more often before a target change and a false positive potentially elicited: the target changes in Experiment 3 could occur from 412 to 2176 ms after RDP onset. The distractor change occurred from 470 to 941 ms after RDP onset so that the distractor changes now occurred more often (about 60% of target-changes were now preceded by distractor changes) and mostly before the saccade. Distractor changes occurred either to the right of the target (to measure post-saccadic retinotopic persistence) or to the left of the target (to measure pre-saccadic predictive shifts). About 6% of trials were catch trials.

## Data analysis

Data processing was done using MATLAB (Mathworks Inc, Natwick, MA), except for the exact test of binomial proportions performed using the Exact package (*Calhoun, 2015*) in R (*R Core Team, 2016*). We detected saccades using a standard velocity-threshold algorithm: onset (and offset) times were determined by when the eye velocity exceeded (and then dropped below) an individualized threshold (set to between 40 and 70° per second, fixed for each subject). This threshold value was set to lie clearly above the peak excursions of the baseline noise in the eye-velocity traces, and the algorithm was validated by visual inspection for each subject. By considering the saccade to have ended when the velocity dropped below a threshold value well above the baseline noise (and when the eye was still moving), our threshold criterion provides a conservative, i.e. early definition of saccadic end-point and therefore a longer estimate of the recovery time for perceptual performance. Our threshold-setting detected the primary saccade close to its end, but excluded post-saccadic dynamic overshoots or glissades (*Bahill et al., 1975*; *Nyström et al., 2013*). Setting a lower threshold and including these small eye-movements led to an even lower estimate of the recovery time of spatial attention (around 20 ms, instead of the 30 ms we report). We only included trials where the subjects made a single saccade to the saccade target, and this saccade was made between 50 ms before and 450 ms after the time when the fixation point jumped. While these limits are arbitrary, they are not critical and our results remain robust for other reasonable choices, consistent with the lack of an effect of saccade latency on performance (*Figure 3*).

Trials with a fixation break were excluded from further analysis. Early responses before the target-change were extremely rare: early responses constituted only 1.2, 1 and 0.7% of trials in Experiments 1, 2 and 3 respectively, even when counting all early responses that were potentially responses to the distractor change in this number. Responses to the distractor change (false-alarms; see Results and discussion) were also extremely rare; we considered all early responses within 800 ms of a distractor change as a response to the distractor. We could therefore exclude trials with early responses as well and simply define performance using the hit-rate (the proportion of target-changes that were correctly detected). We plotted the performance as a function of the time of target-change relative to saccade offset: since the speed and direction-change lasted 2 frames (at a refresh rate of 85 Hz), we used the timing of the second frame to define the time of target-change since this was the conservative choice given our focus on the rapid performance recovery after the saccade. For the pooled analyses (*Figures 2*, *3* and *Figure 2—figure supplement 3A*), we pooled the trials from all subjects and then calculated the mean and 95% Wilson-score confidence intervals (*Brown et al., 2001*) over successive non-overlapping time-bins of the X-axis variable (10 ms in *Figure 2A*, 20 ms in *Figure 2B*, 50 ms in *Figure 3* and 10 ms in *Figure 2—figure supplement 3A*). To estimate the time at which performance recovered to its post-saccadic baseline, we first estimated the baseline performance (proportion of correct trials) from 100 to 500 ms following saccade offset and then compared this value (using Boschloo's exact test of binomial proportions and a one-sided p-value for the peri-saccadic performance being lower than the baseline performance) to the performance in successive non-overlapping 10 ms time-bins from 0 to 100 ms following saccade offset. The starting-point of the first non-significant bin (i.e. p>0.1) was taken as the time of recovery. Using a one-sided p-value and a cutoff of 0.1 are both conservative choices in our situation since they would only increase the estimated time of recovery. Using a cutoff of p>0.05 for non-significance reduced the estimated time of recovery in Experiment 1 (*Figure 2A*) to 20 ms, but did not affect any of the other estimates. Similarly, the use of Boschloo's test also increases the power to detect a significant difference, and is therefore conservative for our purposes (*Berger, 1994*; *Mehrotra et al., 2003*). The time estimated using 10 ms bins was further confirmed with a similar

procedure using 5 ms bins. In all cases (*Figures 2A, B* and *3*), the estimated value was 30 ms, meaning that the performance in the time-bin from both 30 to 35 ms and 30 to 40 ms was not significantly different from baseline. For Experiment 1 (*Figure 2A*), there were at least 48 trials in each 10 ms time-bin from 0 to 40 ms. For the other experiments, the values were: Experiment 2 (*Figure 2B*) – 42 trials for the easy task, and 39 trials for the hard task and Experiment 3 (*Figure 2—figure supplement 3*) – 31 trials. These trial numbers gave us 80% power to detect a reduction to 90% (Experiment 1), 90% (Experiment 2, easy), 81% (Experiment 2, hard) and 83% (Experiment 3) of the baseline value, and the estimated recovery times agreed well with the values one would estimate based on visual inspection of the curves. For the individual subjects (*Figure 2—figure supplements 1–3*), the time-courses appear very similar to the pooled averages. However, formal statistical testing was precluded by the small number of trials in each bin, since the estimates of recovery time based on statistical significance would be shorter than the estimate for the pooled averages (and therefore anti-conservative). We therefore marked the estimated time at which the performance reached 80% of the baseline probability on the individual subject plots in *Figure 2—figure supplements 1 to 3*. This value was calculated via simple linear interpolation and by visual inspection, captures the time-course of recovery quite well. We collected data from a planned number of 8 subjects in Experiment 1. Since the data from the 8 subjects in Experiment 1 showed very similar time-courses, we collected data from a smaller number of 5 and 4 subjects respectively in the additional Experiments (2 and 3).

## Acknowledgements

This work was supported by the Deutsche Forschungsgemeinschaft through the Collaborative Research Center 889 "Cellular Mechanisms of Sensory Processing" to ST (Project C04), and by the Germany-Israeli Foundation for Scientific Research and Development Grant No. 1108–79.1/2010.

## Additional information

### Funding

| Funder | Grant reference number | Author |
| --- | --- | --- |
| Deutsche Forschungsgemeinschaft | Collaborative Research Center 889 | Stefan Treue |
| German-Israeli Foundation for Scientific Research and Development | 1108-79.1/2010 | Stefan Treue |

The funders had no role in study design, data collection and interpretation, or the decision to submit the work for publication.

### Author contributions

TY, Conception and design, Acquisition of data, Analysis and interpretation of data, Drafting or revising the article; MK, Acquisition of data, Analysis and interpretation of data; ST, Analysis and interpretation of data, Drafting or revising the article, Contributed unpublished essential data or reagents; BSK, Conception and design, Analysis and interpretation of data, Drafting or revising the article

### Author ORCIDs

Tao Yao, http://orcid.org/0000-0003-2060-690X
B Suresh Krishna, http://orcid.org/0000-0002-0383-054X

### Ethics

Human subjects: The study was performed in accordance with institutional guidelines for experiments with humans, adhered to the principles of the Declaration of Helsinki and was approved by the Ethics Committee of the Georg-Elias-Müller-Institute of Psychology, University of Göttingen. Each subject gave informed written consent prior to participating in the study.

## Additional files

**Supplementary files**
• Supplementary file 1. The false-positive rate shows no effect of either pre-saccadic predictive shifts or post-saccadic retinotopic persistence. (A) Experiments 1 and 2. False-positive rate from 0 to 150 ms after saccade offset shows no effect of post-saccadic retinotopic attentional persistence. (B) The false-positive rate in Experiment 3 shows no effect of either pre-saccadic predictive shifts or post-saccadic retinotopic persistence.

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
