## [Decision Letter]

Thank you for submitting your article "Visual attention is available at a task-relevant location rapidly after a saccade" for consideration by *eLife*. Your article has been favorably evaluated by David Van Essen (Senior Editor) and two reviewers, one of whom, Jack Gallant, is a member of our Board of Reviewing Editors. The reviewers have opted to remain anonymous. The reviewers have discussed the reviews with one another and the Reviewing Editor has drafted this decision to help you prepare a revised submission.

General assessment and central conclusions:

The reviewers agree that your paper is very interesting, worthwhile and potentially suitable for publication in *eLife*. The basic design of the experiment is elegant, the controls are reasonable and the results seem to be fairly clear. The experiments demonstrate that spatial remapping to the correct target is timed appropriately relative to the saccade, and they provide precise time estimates for this process. These novel results point to a surprisingly quick recovery of attention after saccades, suggesting that both eye movements and attention work in concert to optimize perception.

Essential revisions:

Although the results appear to be solid as far as they go, additional behavioral data will be required in order to fill in some missing controls and to address potential confounds.

1) Behavioral controls should be included that vary target location across trials, and in which the distractors are not present. The authors assert that two critical differences between the current study and previous efforts are (1) target position known and was blocked across trials, and (2) irrelevant distractors were present at known locations. If these differences are indeed key to the results, then the results should change (becoming more similar to previous studies) if these differences are removed.

2) In the eye movement task the fixation point jump timing and location were deterministic. This might have influenced the results somehow (for example, if subjects loaded up the eye movement vector beforehand and ignored the fixation point.) A behavioral control should be included that evaluates this issue.

3) Subjects had to maintain their gaze within 2 degrees of the fixation point, which is a rather large eye window to use for an experiment that evaluates the relationship between eye position and attention. Further analysis and discussion should be provided to demonstrate that this large eye window did not unduly influence the results (for example, it should be possible to show whether the results are the same when the data analysis is restricted to the trials where subjects fixated well).

---

## [Author Response]

*[…] Essential revisions:*

*Although the results appear to be solid as far as they go, additional behavioral data will be required in order to fill in some missing controls and to address potential confounds.*

*1) Behavioral controls should be included that vary target location across trials, and in which the distractors are not present. The authors assert that two critical differences between the current study and previous efforts are (1) target position known and was blocked across trials, and (2) irrelevant distractors were present at known locations. If these differences are indeed key to the results, then the results should change (becoming more similar to previous studies) if these differences are removed.*

Our primary conclusion in this manuscript is about the availability of spatial attention at the attended location soon after the saccade, and this conclusion is not in conflict with the results from any of the previous studies, even though they did not measure this explicitly. Additionally, as a secondary finding, we report that some of the attentional effects reported in previous studies (where peri-saccadic perceptual benefits were found at locations other than the target location) were not found in our task: subjects almost never made false-positive responses to distractor changes. In our original manuscript, we suggested two possible explanations for why a difference in task-design may have led to this difference in results. However, unfortunately, it seems that our discussion of this point was not sufficiently clear and this has led to some misunderstanding.

We pointed out that our study differed from previous studies in one critical way: there was only one potential target location *within* each trial that the subjects had to attend to, and as a consequence, all other locations used *within a given trial* were distractors that the subjects had to ignore. Based on this, we suggested two possibilities: a) The subjects may have utilized a different (possibly more diffuse/flexible) attentional process with different dynamics in the previous tasks. We mentioned this because it is logically possible. However, this explanation is also not directly testable, at least using psychophysical methods. b) We also suggested that alternatively, because the target could only appear at one location *in a given trial* in our task, the subjects could set up a spatial filter that strongly down-weighted information from irrelevant spatial locations. As a result of this down-weighting, if the attentional improvements in the previous tasks were based on signal enhancement, these improvements (though present) would still not count in the decision process in our task. If we tested a condition where there are no distractors (as you recommend above) and every stimulus is a potential target, then the task reduces to a simple measurement of saccadic suppression with no attentional component, since the subjects are now instructed to respond to any stimulus change they see on the screen. To introduce an attentional component, there are two general possibilities, and these have been used in previous studies. First, one could introduce an exogenous (Jonikaitis et al., 2013) or endogenous cue (e.g. by making the target more probable at one location compared to the others). Second, as an alternative, one could create a dual-task scenario where the subject performs an additional secondary task, like a double-step saccade (Rolfs et al., 2011) or a spatial memory task (Golomb et al., 2008; Golomb et al., 2010a) or tracking a moving color patch (Szinte et al., 2015). The dual task serves the purpose of focusing attention on one stimulus, even though the target change could appear at all stimuli (under the assumption that there is a common attentional resource that is focused similarly for both tasks).

In all these scenarios, our control experiment would become a replication of the general experimental paradigm from one of the previous studies. We therefore discussed our results regarding the absence of false-positive responses by accepting the results from the previous studies and contrasting those results with ours, instead of attempting to replicate the previous results. We did this because: a) Our primary conclusion is about the availability of spatial attention at the attended location soon after the saccade, and this is not in conflict with previous studies. The control experiments discussed above address a secondary and relatively minor aspect of our conclusions. b) The control experiments discussed above are not “clean”, in the sense that they introduce other task differences that complicate their interpretation in terms of the underlying model. For example, all the proposed controls presumably increase the task-difficulty, and also introduce the complication of a dual-task. Thus, while the proposed control modifications may be intended to broaden the attentional weighting, they may also change other aspects of the model (like attentional strength or dynamics or threshold settings) making their outcome hard to interpret. Independent of whether our replication controls agreed or disagreed with the results of previous studies, the basic attentional model we consider (Lu and Dosher, 1998; Eckstein et al., 2009) is sufficiently flexible to accommodate any differences in multiple ways. As a result, firm conclusions can only be made after a detailed model-testing exercise which is well beyond the scope of this Short Report.

We however, appreciate the reviewer’s concern about our making an untested suggestion regarding why our secondary conclusion differs from previous studies. We have therefore modified the manuscript to better reflect that there are multiple possible explanations. We have removed the reference to “signal enhancement” versus “attentional selection” from the Abstract and Introduction, and modified the Discussion section as well. We hope that these changes will address the reviewers’ concerns, in the light of our additional explanation above. Also, for completeness, we hope that it is now clear that the other control experiment that the reviewer desired, where the target location is varied *across trials* does not test a hypothesis that is relevant to our conclusions (as the clarification regarding *within trial* and *across trial* variations in target location above indicates). This control experiment instead tests how well subjects can switch their attentional set (i.e. the locus of spatial attention) across successive trials – a topic that is of course interesting, but represents a separate line of enquiry.

*2) In the eye movement task the fixation point jump timing and location were deterministic. This might have influenced the results somehow (for example, if subjects loaded up the eye movement vector beforehand and ignored the fixation point.) A behavioral control should be included that evaluates this issue.*

Once again, it appears that we were not sufficiently clear on this point, since our original manuscript had just one sentence in the Methods section addressing this issue (“The use of a predictable time at which the fixation point jumped was advantageous because subjects could focus their attention better on the target RDP without worrying about the temporal uncertainty about when the fixation point would jump”). It was in fact important for the design of our experiment that the subjects paid as little attention to the fixation point as possible and devoted attention to the specified target location. The goal of our experiment was to ask how well the system could remap attentional resources across a saccade, and therefore it was important to minimize the costs imposed by the diversion/splitting of attentional resources to exogenous visual events (like detecting the disappearance of the fixation point and the appearance/location of the saccade target). We now make this more clear by adding this sentence at the end of the first paragraph of the Results and Discussion section: “By using a fixed timing and location for the fixation point jump, we could isolate the dynamics of this attentional remapping process and minimize its interaction with the dynamics of attentional allocation to other exogenous visual events: we therefore made the saccade spatially and temporally predictable by having the fixation point jump at the same time and to the same location on each trial so that the subject could focus on the target pattern.”

The proposed experiment actually asks a new question about how the system performs when the target motion patch and other new, sudden-onset visual events on the screen (like the disappearance of the fixation point and the appearance of the saccade target) compete for attention. The measured performance under these circumstances would reflect both attentional remapping as well as the dynamics of attentional shifts and/or splitting between the fixation point, the saccade target and the target motion patch. It would also reflect an interaction between endogenous attention (to the attended target) and exogenous attention (to the disappearance of the fixation point and appearance of the saccade target). While this is of course interesting, we think that this represents an extension of the work we report here, rather than a control experiment.

*3) Subjects had to maintain their gaze within 2 degrees of the fixation point, which is a rather large eye window to use for an experiment that evaluates the relationship between eye position and attention. Further analysis and discussion should be provided to demonstrate that this large eye window did not unduly influence the results (for example, it should be possible to show whether the results are the same when the data analysis is restricted to the trials where subjects fixated well).*

We have now included a supplementary figure (Figure 2—figure supplement 5) that only includes trials where the eye did not deviate by more than 0.5 degrees from the median eye-position (calculated within each trial). As the figure shows, our results are very robust to such a sub-selection of trials. The procedure of using the median eye-position removes the effect of across-trial drifts in calibration is based on the standard assumption that subjects do not fixate a small stimulus para-foveally. We also noticed and fixed some minor (and inconsequential) errors in the way we implemented the fixation window checks in our analysis code, and as a result, all our figures have changed (but almost unnoticeably) and we have updated some numbers in the manuscript.

We would also like to clarify that our original methodology was not lax compared to the standards in the field. Five of the six main previous studies (regarding the relationship of eye movements to attention) that we refer to here used a window of at least 1.8 degrees: one used a window of 1.8 degrees (Golomb et al., 2010b), another three used a window of 2 degrees (Golomb et al., 2008; Golomb et al., 2010a; Rolfs et al., 2011), while the fifth used a window of 2.5 degrees (Szinte et al., 2015). The sixth study (Jonikaitis et al., 2013) does not report the window size used, but does report that only “trials in which the saccade landed within a 2° radius around its goal” were analyzed. Further, in our experience, the main reason labs use a window around this size is because of across-trial drifts in calibration, not within-trial variation in eye-position during fixation (at least with small fixation stimuli).